# CONTEXTUAL TRANSFORMATION NETWORKS FOR ONLINE CONTINUAL LEARNING

**Quang Pham[1], Chenghao Liu[2], Doyen Sahoo[2], Steven C.H. Hoi [1,2]**
[1] Singapore Management University
`hqpham.2017@smu.edu.sg`
[2] Salesforce Research Asia
`{chenghao.liu, dsahoo, shoi}@salesforce.com`

## ABSTRACT

Continual learning methods with fixed architectures rely on a single network to learn models that can perform well on all tasks. As a result, they often only accommodate common features of those tasks but neglect each task's specific features. On the other hand, dynamic architecture methods can have a separate network for each task, but they are too expensive to train and not scalable in practice, especially in online settings. To address this problem, we propose a novel online continual learning method named "Contextual Transformation Networks" (CTN) to efficiently model the *task-specific features* while enjoying neglectable complexity overhead compared to other fixed architecture methods. Moreover, inspired by the Complementary Learning Systems (CLS) theory, we propose a novel dual memory design and an objective to train CTN that can address both catastrophic forgetting and knowledge transfer simultaneously. Our extensive experiments show that CTN is competitive with a large scale dynamic architecture network and consistently outperforms other fixed architecture methods under the same standard backbone. Our implementation can be found at `https://github.com/phquang/Contextual-Transformation-Network`.

## 1 INTRODUCTION

Continual learning is a promising framework towards building AI models that can learn continuously through time, acquire new knowledge while being able to perform its already learned skills (French, 1999; 1992; Parisi et al., 2019; Ring, 1997). On top of that, online continual learning is particularly interesting because it resembles the real world and the model has to quickly obtain new knowledge on the fly by levering its learned skills. This problem is important for deep neural networks because optimizing them in the online setting has been shown to be challenging (Sahoo et al., 2018; Aljundi et al., 2019a). Moreover, while it is crucial to obtain new information, the model must be able to perform its acquired skills. Balancing between preventing catastrophic forgetting and facilitating knowledge transfer is imperative when learning on a stream of tasks, which is ubiquitous in realistic scenarios. Thus, in this work, we focus on the continual learning setting in an online learning fashion, where both tasks and data of each task arrive sequentially (Lopez-Paz & Ranzato, 2017).

In the literature, fixed architecture methods employ a shared feature extractor and a set of classifiers, one for each task (Lopez-Paz & Ranzato, 2017; Chaudhry et al., 2019a;b; Aljundi et al., 2019a). Although using a shared feature extractor has achieved promising results, the common and global features are rather generic and not well-tailored towards each specific task. This problem is even more severe when old data are limited while learning new tasks. As a result, the common feature extractor loses its ability to extract previous tasks' features, resulting in catastrophic forgetting. On the other hand, while dynamic architecture methods such as Rusu et al. (2016); Li et al. (2019); Xu & Zhu (2018) alleviate this problem by having a separate network for each task, they suffer from the unbounded growth of the parameters. Moreover, the subnetworks' design is not trivial and requires extensive resource usage (Rusu et al., 2016; Li et al., 2019), which is not practical in many applications. These limitations motivated us to develop a novel method that can facilitate continual learning with a fixed architecture by modeling the task-specific features.

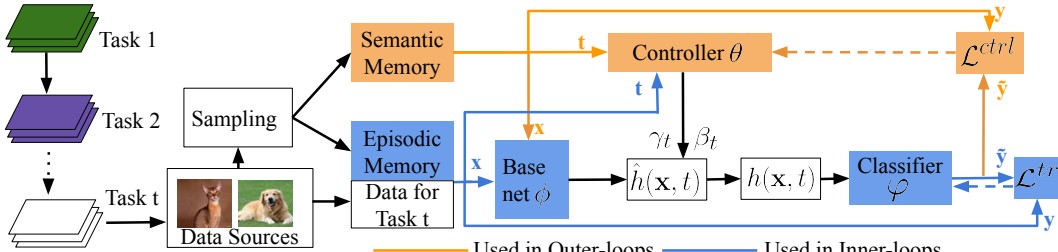

Figure 1: Overview of the Contextual Transformation Networks (CTN). CTN consists of a controller $\theta$ that modifies the features of the base model $\phi$. The base model is trained using experience replay on the episodic memory while the controller is trained to generalize to the semantic memory, which addresses both alleviating forgetting and facilitating knowledge transfer. Best viewed in colors.

To achieve this goal, we first revisit a popular result in learning multiple tasks that each task's features are centered around a common vector (Evgeniou & Pontil, 2004; Aytar & Zisserman, 2011; Pentina & Lampert, 2014; Liu et al., 2019b). This result motivates us to develop a novel framework of Contextual Transformation Networks (CTN), which consists of a base network that learns the common features of a given input and a controller that efficiently transforms the common features to become task-specific, given a task identifier. While one can train CTN using experience replay, it does not explicitly aim at achieving a good trade-off between stability and plasticity. Therefore, we propose a novel dual memory system and a learning method that encapsulate alleviating forgetting and facilitating knowledge transfer simultaneously. Particularly, we propose two distinct memories: the *episodic memory* and the *semantic memory* associated with the base model and the controller, respectively. Then, the base model is trained by experience replay on the episodic memory while the controllers is trained to learn task-specific features that can generalize to the semantic memory. As a result, CTN achieves a good trade-off between preventing catastrophic forgetting and facilitating knowledge transfer because the task-specific features can generalize well to all past and current tasks. Figure 1 gives an overview of the proposed Contextual Transformation Network (CTN).

Interestingly, the designs of our CTN and dual memory are partially related to the Complementary Learning Systems (CLS) theory in neuroscience (McClelland et al., 1995; Kumaran et al., 2016). Particularly, the controller acts as a *neocortex* that learns the structured knowledge of each task. In contrast, the base model acts as a *hippocampus* that performs rapid learning to acquire new information from the current task's training data. Following the naming convention of memory in neuroscience, our CTN is equipped with two replay memory types. (i) the **episodic memory** (associated with the hippocampus) caches a small amount of past tasks' training data, which will be replayed when training the base networks. (ii) the **semantic memory** (associated with the neocortex) stores another distinct set of old data only used to train the controller such that the task-specific features can generalize well across tasks. Moreover, the CLS theory also suggests that the interplay between the neocortex and the hippocampus attributes to the ability to recall knowledge and generalize to novel experiences (Kumaran & McClelland, 2012). Our proposed learning approach closely characterizes such properties: the base model focuses on acquiring new knowledge from the current task while the controller uses the base model's knowledge to generalize to novel samples.

In summary, our work makes the following contributions. First, we propose CTN, a novel continual learning method that can model task-specific features while enjoying neglectable complexity overhead compared to fixed architecture methods (please refer to Table 4). Second, we propose a novel objective that can improve the trade-off between alleviating forgetting and facilitating knowledge transfer to train CTN. Third, we conduct extensive experiments on continual learning benchmarks to demonstrate the efficacy of CTN compared to a suite of baselines. Finally, we provide a comprehensive analysis to investigate the complementarity of each CTN's component.

## 2 METHOD

**Notations.** We denote $\phi$ as parameter of the base model that extracts global features from the input and $\theta$ as the parameter of the controller which modifies the features from $\phi$ given a task identifier $t$. The task identifier can be a set of semantic attributes about objects of that task (Lampert et al., 2009) or simply an index of the task, which we use in this work as a one-hot vector. A prediction is given as $g_{\varphi_t}(h_{\phi,\theta}(\boldsymbol{x},t))$, where $g_{\varphi_t}(\cdot)$ is the task $\mathcal{T}_t$'s classifier with parameter $\varphi_t$ such as a fully

connected layer with softmax activation. And $h_{\phi,\theta}(\boldsymbol{x}, t)$ is the final feature after transformed by the controller. We denote $D_t^{tr}$ as the training data of task $\mathcal{T}_t$, $\mathcal{M}_t^{em}$ and $\mathcal{M}_t^{sm}$ as the *episodic memory* and the *semantic memory* of task $\mathcal{T}_t$ respectively. The episodic memory and semantic memory maintains two distinct sets of data obtained from task $\mathcal{T}_t$. The episodic memory of task $\mathcal{T}_1, \ldots, \mathcal{T}_{t-1}$ is denoted as $\mathcal{M}_{<t}^{em}$; similarly, $\mathcal{M}_{<t}^{sm}$ denotes the semantic memory of the first $t-1$ tasks.

**Remark.** Both the $\mathcal{M}_t^{em}$ and $\mathcal{M}_t^{sm}$ are obtained from $D_t^{tr}$ through the learner's internal memory management strategy and contains distinct samples from each other such that their combined sizes do not exceed a pre-defined budget.

## 2.1 LEARNING TASK-SPECIFIC FEATURES FOR CONTINUAL LEARNING

Given a backbone network, one can implement the task-specific features by employing a set of task-specific filters and applying them to the backbone's output. However, this trivial approach is not scalable, even for small networks. In the worst case, it results in storing an additional network per task, which violates the fixed architecture constraint. Since we want to obtain task-specific features with minimal parameter overhead, we propose to use a feature-wise transformation (Perez et al., 2018) to efficiently extract the task-specific features $\tilde{h}(\boldsymbol{x}, t)$ from the common features $\hat{h}(\boldsymbol{x})$ as follows:

$$\tilde{h}(\boldsymbol{x}; t) = \frac{\gamma_t}{\|\gamma_t\|_2} \otimes \hat{h}(\boldsymbol{x}) + \frac{\beta_t}{\|\beta_t\|_2} \text{ and } \{\gamma_t, \beta_t\} = c_{\boldsymbol{\theta}}(t), \tag{1}$$

where $\otimes$ denotes the element-wise multiplication operator, and $c_{\boldsymbol{\theta}}(t)$ is the controller implemented as a linear layer with parameter $\boldsymbol{\theta}$ that predicts the transformation coefficients $\{\boldsymbol{\gamma}_t, \beta_t\}$ given the task identifier $t$. Since the task identifiers are one-hot vectors, which are sparse and make training the controller difficult, we also introduce an embedding layer to map the task identifiers to dense, low dimensional vectors. For simplicity, we will use $\boldsymbol{\theta}$ to refer to both the embedding and the linear layer parameters. In addition, instead of storing a set of coefficients $\{\gamma_t, \beta_t\}$ for each task, we only need a fixed set of parameter $\boldsymbol{\theta}$ to predict these coefficients, which results in the fixed parameters in the controller. The coefficients $\{\boldsymbol{\gamma}_t, \beta_t\}$ are $\ell_2$-normalized and then transforms the common features $\hat{h}(\boldsymbol{x}, t)$ to become task-specific features $\tilde{h}(\boldsymbol{x}; t)$. Finally, both feature types are combined by a residual connection before passing to the corresponding classifier $g_t(\cdot)$ to make the final prediction:

$$g_{\boldsymbol{\varphi}_t}(\sigma(h(\boldsymbol{x}, t))), \text{ and } h(\boldsymbol{x}, t) = \hat{h}(\boldsymbol{x}, t) + \tilde{h}(\boldsymbol{x}, t), \tag{2}$$

where $\sigma(\cdot)$ is a nonlinear activation function such as ReLU. Importantly, when the task-specific features are removed, i.e., $\tilde{h}(\boldsymbol{x}, t) = 0$, Eq. 2 reduces to the traditional experience replay. Lastly, for each incoming task, CTN has to allocate a new classifier, which is the same for all continual learning methods, and a new embedding vector, which is usually low dimensional, e.g. 32 or 64. Therefore, CTN enjoys almost the same parameter growth as existing continual learning methods.

## 2.2 TRAINING THE CONTROLLER

While one can train CTN with experience replay (ER), it does not explicitly address the trade-off between facilitating knowledge transfer and alleviating catastrophic forgetting. This motivates us to develop a novel training method that can simultaneously address both problems by leveraging the controller's task-specific features. First, we introduce a dual memory system consisting of the semantic memory $\mathcal{M}_t^{sm}$ associated with the controller and the episodic memory $\mathcal{M}_t^{em}$ associated with the base model. We propose to train only the base model using experience replay with the episodic memory to obtain new knowledge from incoming tasks. The controller is also trained so that the task-specific features can generalize to unseen samples to the base model stored in the semantic memory. As a result, the task-specific features can generalize to both previous and current tasks, which simultaneously encapsulate both alleviating forgetting and facilitating knowledge transfer. Formally, given the current batch of data for task $\mathcal{T}_t$ as $\mathcal{B}_t$, the training of CTN can be formulated as the following bilevel optimization problem (Colson et al., 2007):

$$\begin{aligned} \text{Outer problem:} \quad &\min_{\boldsymbol{\theta}} \mathcal{L}^{ctrl}(\{\boldsymbol{\phi}^*, \boldsymbol{\theta}\}; \mathcal{M}_{<t+1}^{sm}) \\ \text{Inner problem:} \quad \text{s.t} \quad &\boldsymbol{\phi}^* = \arg\min_{\boldsymbol{\phi}} \mathcal{L}^{tr}(\{\boldsymbol{\phi}, \boldsymbol{\theta}\}, \mathcal{B}_t \cup \mathcal{M}_{<t}^{em}), \end{aligned} \tag{3}$$

where $\boldsymbol{\phi}^*$ denotes the optimal base model corresponding to the current controller $\boldsymbol{\theta}$. Since every CTN's prediction always involves both the controller and the base model, we use

$\mathcal{L}^{tr}(\{\phi, \theta\}, \mathcal{B}_t \cup \mathcal{M}_{<t}^{em})$ to denote the training loss of the pair $\{\phi, \theta\}$ on the data $\mathcal{B}_t \cup \mathcal{M}_{<t}^{em}$. Similarly, $L^{ctrl}(\cdot)$ denotes the controller's loss. For simplicity, we omitted the dependency of the the loss on the classifiers' parameters and imply that the classifiers are jointly updated with the base model. Since we do not know the optimal transformation coefficients of any task, the controller is trained to minimize the classification loss of the samples via $\phi$. We implement both the training and controller's losses as the cross-entropy loss. Notably, Eq. 3 characterizes two nested optimization problems: the outer problem, which trains the controller to generalize, and each controller parameter $\theta$ parameterizes an inner problem that trains the base model to acquire new knowledge via experience replay. Moreover, only $\phi$ is trained in the inner problem, while only $\theta$ is updated in the outer problem.

The bilevel optimization objective such as Eq. 3 has been successfully applied in other machine learning disciplines such as hyperparameter optimization, meta learning (Franceschi et al., 2018; Finn et al., 2017), and AutoML (Liu et al., 2019a). In this work, we extend this framework to continual learning to train the controller. However, unlike existing works (Franceschi et al., 2018; Finn et al., 2017; Liu et al., 2019a), our Eq. 3 has to be solved incrementally when a new data sample arrives. Therefore, we consider Eq. 3 as an online learning problem and optimize it using the *follow the leader* principle (Hannan, 1957). Particularly, we relax the optimal solutions of both the inner and outer problems to be solutions from a few gradient steps. When a new training data arrives, we first train the base model $\phi$ using experience replay for a few SGD steps with an inner learning rate $\alpha$, each of which is implemented as:

$$\phi \leftarrow \phi - \alpha \nabla_\phi \mathcal{L}^{tr}(\{\phi, \theta\}, \mathcal{B}_t \cup \mathcal{M}_{<t}^{em}), \tag{4}$$

Then, we optimize the controller $\theta$ such that it can improve $\phi$'s performance on the semantic memory:

$$\theta \leftarrow \theta - \beta \nabla_\theta \mathcal{L}^{ctrl}(\{\phi, \theta\}, \mathcal{M}_{<t+1}^{sm}), \tag{5}$$

where $\beta$ is the outer learning rate. As a result, Eq. 3 is implemented as an alternative update procedure involving several outer updates to train $\theta$, each of which includes an inner update to train $\phi$. Moreover, performing several updates per incoming sample does not violate the online assumption since we will not revisit that sample in the future, unless it is stored in the memories.

## 2.3 Training the base network

Despite using task-specific features, the base network may still forget previous tasks because of the small episodic memory. To further alleviate catastrophic forgetting in $\phi$, we regularize the training loss $\mathcal{L}^{tr}(\cdot)$ with a behavioral cloning (BC) strategy based on knowledge distillation (Hinton et al., 2015; van de Ven & Tolias, 2018). Let $\hat{y}$ be the logits of the model's prediction before the softmax layer $\pi(\cdot)$, we regularize the training loss on the episodic memory data in Eq. 4 as:

$$\mathcal{L}^{tr}(\{\phi, \theta\}, (\boldsymbol{x}, y, k)) = \mathcal{L}(\pi(\hat{y}), y) + \lambda D_{\mathrm{KL}}\left(\pi\left(\frac{\hat{y}}{\tau}\right) \middle\| \pi\left(\frac{\hat{y}_k}{\tau}\right)\right), \tag{6}$$

where $\lambda$ is the trade-off parameter, $\tau$ is the softmax's temperature, and $\hat{y}_k$ is a snapshot of the model prediction on the sample $(\boldsymbol{x}, k)$ at the end of task $\mathcal{T}_k$. While the behavioral cloning strategy requires storing $\hat{y}_k$, the memory increase is minimal since $\hat{y}_k$ is a vector with dimension bounded by the total classes, which is much smaller than the image $\boldsymbol{x}$ dimension. Importantly, the behavioural cloning strategy is used to alleviate catastrophic forgetting, which only happens in the base model, not the controller. Particularly, the controller's inputs are task identifiers such as one-hot vectors, which are fully available during learning. In summary, our episodic memory stores the input image $\boldsymbol{x}$, its corresponding label $y$ and the soft label $\hat{y}$, while the semantic memory stores the input-label pair $\boldsymbol{x}, y$.

## 3 Related Work

### 3.1 Continual Learning

Prior works in continual learning can be grouped into three main categories: (1) regularization methods, (2) episodic memory based methods, and (3) dynamic architecture methods.

**Regularization** approaches (Kirkpatrick et al., 2017; Zenke et al., 2017; Aljundi et al., 2018; Ritter et al., 2018) penalize the changes of important parameters to previous tasks using a variant of

knowledge distillation (Li & Hoiem, 2017) or via a quadratic constraints. However, such methods usually isolate parameters or find a common solution to all tasks, limiting the model's capacity.

**Episodic memory** based approaches store a small amount of data from previous tasks and interleave it with data from the current task. Old data can be used as a constraint to optimize the model Lopez-Paz & Ranzato (2017); Chaudhry et al. (2018), representation learning Rebuffi et al. (2017b), or simply just perform experience replay (ER) Chaudhry et al. (2019b); Aljundi et al. (2019a); Rolnick et al. (2019); van de Ven & Tolias (2018). While regularization and episodic memory-based methods have achieved promising results, they only use a shared feature extractor. Moreover, they do not consider the goal of improving the generalizability across tasks, which CTN explicitly addresses via the proposed bi-level optimization with the dual memory design.

**Dynamic architecture** approaches address catastrophic forgetting by having a subnetwork for each task (Rusu et al., 2016; Serra et al., 2018; von Oswald et al., 2020) or being able to grow its structure over time (Yoon et al., 2018; Li et al., 2019; Xu & Zhu, 2018; Hung et al., 2019). Such methods approximate training a full, separate network per task by reducing the number of additional parameters. However, most of them require growing the backbone network during training (Rusu et al., 2016; Yoon et al., 2018; Xu & Zhu, 2018; Li et al., 2019) or extensive resource usage (Rusu et al., 2016; Li et al., 2019), which is not scalable and undesirable for many applications. Notably, the idea of conditioning on the task identifiers were explored in Serra et al. (2018); von Oswald et al. (2020). However, Serra et al. (2018) uses the task identifiers to gate the network's activations, which limits the representation capability. On the other hand, von Oswald et al. (2020) employs a hypernetwork (Ha et al., 2017) to generate a whole prediction network for each task and catastrophic forgetting is avoided by performing experience replay in the hypernetwork's output space. However, this approach requires storing the hypernetwork's output for each task, which is equivalent to a prediction network's parameter. Therefore, while Serra et al. (2018); von Oswald et al. (2020) have achieved promising results, they requires larger memory and might not be suitable for the online setting.

### 3.2 FEATURE-WISE TRANSFORMATION

Early works in Bertinetto et al. (2016); Rebuffi et al. (2017a) showed that instead of using a task-specific network on the input, one can employ a set of $1 \times 1$ filters to extract the task-specific from the common features. However, such approaches still require a *quadratic* complexity overhead in the number of channels, which can be expensive. Another compelling solution is the feature-wise transformation, FiLM (Perez et al., 2018), which only requires a linear complexity. Thanks to its efficiency, FiLM has been successfully applied in many problems, including meta learning (Requeima et al., 2019; Zintgraf et al., 2019), visual reasoning (Perez et al., 2018), and others fields (Dumoulin et al., 2018) with remarkable success. Notably, CNAPs (Requeima et al., 2019) proposed an adaptation network to generate the FiLM's parameters and quickly adapt to new tasks. CNAPs has showed promising results when having access to a large amount of tasks to pre-train the common features. However, this setting is different from continual learning where the learner has to obtain new knowledge on the fly. Therefore, CNAPs are principally differs from CTN in that CNAPs assume having access to a well-pretrained knowledge source and uses FiLM to quickly adapt this knowledge to a new task. On the other hand, CTNs use FiLM to accelerate the knowledge acquisition when learning progressively. Lastly, we emphasize that the CTN's design is general. If more budget is allowed, the proposed CTN is readily compatible with the aforementioned feature transformation methods such as Rebuffi et al. (2017a) by adjusting the controller's output dimension.

### 3.3 META LEARNING

Meta learning (Schmidhuber, 1987), also learning to learn, refers to a learning paradigm where an algorithm learns to improve the performance of another algorithm. Our CTN design is related to such learning to learn architectures where the controller is trained to improve the base model's performance. Importantly, we note that there exist other continual learning variants that intersect with meta learning, such as meta-continual learning (Javed & White, 2019) and continual-meta learning (He et al., 2019; Caccia et al., 2020). However, they consider *different* goals and problem settings, such as meta pre-training (Javed & White, 2019) or rapid recovering the performance at test time given a finetuning step before inference is allowed (He et al., 2019), which is *not* the conventional online continual learning problem (Lopez-Paz & Ranzato, 2017) we focus in this study.

Table 1: Evaluation metrics on continual learning benchmarks considered. All methods use the same backbone network and 50 memory slots per task, [*] denotes a dynamic architecture method that has a separate network per task

| Method | pMNIST | | | CORe50 | | |
|---|---|---|---|---|---|---|
| | ACC(↑) | FM(↓) | LA(↑) | ACC(↑) | FM(↓) | LA(↑) |
| GEM | 74.84±0.95 | 8.57±0.33 | 81.74±0.77 | 42.56±0.86 | 7.36±0.90 | 46.84±2.22 |
| AGEM | 68.67±0.71 | 13.98±0.68 | 81.54±0.25 | 40.28±3.15 | 11.08±4.01 | 48.68±1.51 |
| MER | 76.59±0.74 | 6.88±0.59 | 82.09±0.33 | 39.28±1.25 | 9.08±1.25 | 45.52±0.96 |
| ER-Ring | 76.02±0.59 | 8.57±0.33 | 83.69±0.44 | 41.72±1.30 | 9.10±0.80 | 48.18±0.81 |
| MIR | 76.58±0.10 | 8.34±0.11 | 83.57±0.07 | 43.50±1.92 | 6.14±0.91 | 45.98±1.14 |
| CTN (ours) | **79.01±0.65** | **6.69±0.51** | **85.11±0.45** | **54.17±0.85** | **5.50±1.01** | **55.32±0.34** |
| Independent[*] | 81.05±0.29 | 0.00 | 81.05±0.29 | 53.54±1.10 | 0.00 | 53.54±1.10 |
| Offline | 84.95±0.95 | - | - | 58.69±0.41 | - | - |

| Method | Split CIFAR | | | Split miniIMN | | |
|---|---|---|---|---|---|---|
| | ACC(↑) | FM(↓) | LA(↑) | ACC(↑) | FM(↓) | LA(↑) |
| GEM | 57.77±0.86 | 10.93±1.03 | 66.45±0.06 | 55.04±1.88 | 7.81±1.70 | 60.13±1.36 |
| AGEM | 58.27±0.86 | 8.76±0.67 | 66.12±1.17 | 51.14±2.16 | 6.99±1.96 | 55.11±0.76 |
| MER | 61.32±0.86 | 11.90±0.86 | 72.51±0.41 | 57.94±1.08 | 8.98±0.79 | 66.11±0.76 |
| ER-Ring | 61.36±1.01 | 7.20±0.72 | 67.05±1.08 | 53.43±1.18 | 11.21±1.35 | 63.46±1.05 |
| MIR | 63.37±1.99 | 10.53±1.63 | 73.27±0.77 | 51.97±1.58 | 10.37±2.72 | 60.63±3.43 |
| CTN (ours) | **67.65±0.43** | **6.33±0.70** | **73.43±0.45** | **65.82±0.59** | **3.02±1.13** | **67.43±1.37** |
| Independent[*] | 67.21±0.51 | 0.00 | 67.21±0.51 | 65.85±0.98 | 0.00 | 65.85±0.98 |
| Offline | 74.11±0.66 | - | - | 71.15±2.95 | - | - |

Meta learning has been an appealing solution to learn a good initialization from a large amount of tasks (Finn et al., 2017), even in an online manner: Online Meta Learning (OML) (Finn et al., 2019). However, we emphasize that OML fundamentally *differs* from our CTN in two aspects. First, OML requires all data of previous tasks and aims to improve the performance of future tasks, which is different from continual learning. Second, OML learns an initialization and requires finetuning at test time, which is not practical, especially when testing on learned tasks. In contrast, CTN is a continual learning method that maximizes the performance of the current task as well as all previous tasks. Moreover, CTN can make a prediction at any time without requiring an additional finetuning step.

## 4 EXPERIMENTS

### 4.1 BENCHMARK DATASETS AND BASELINES

We consider four continual learning benchmarks in our experiments. **Permuted MNIST (pMNIST)** (Lopez-Paz & Ranzato, 2017): each task is a random but fixed permutation of the original MNIST. We generate 23 tasks with 1,000 images for training and the testing set has the same amount of images as in the original MNIST data. **Split CIFAR-100 (Split CIFAR)** (Lopez-Paz & Ranzato, 2017) is constructed by splitting the CIFAR100 (Krizhevsky & Hinton, 2009) dataset into 20 tasks, each of which contains 5 different classes sampled without replacement from the total of 100 classes. **Split Mini ImageNet (Split miniIMN)** (Chaudhry et al., 2019a), similarly, we split the miniIMN dataset (Vinyals et al., 2016) into 20 disjoint tasks. Finally, we consider the **CORe50** benchmark by constructing a sequence of 10 tasks using the original CORe50 dataset (Lomonaco & Maltoni, 2017).

Throughout the experiments, we compare CTN with a suite of baselines: GEM (Lopez-Paz & Ranzato, 2017), AGEM (Chaudhry et al., 2019a), MER (Riemer et al., 2019), ER-Ring (Chaudhry et al., 2019b), and MIR (Aljundi et al., 2019a). We also consider the *independent* model (Lopez-Paz & Ranzato, 2017), a dynamic architecture method that maintains a separate network for each task, and each has the **same number of parameters** as other baselines. While the independent model is unrealistic, it is highly competitive and was used as an upper bound of a state-of-the-art dynamic architecture method in Hung et al. (2019). Finally, we include the *Offline* model, which does not follow the continual learning setting and performs multitask training on all tasks' data. Due to space constraints, we provide the results of less competitive methods in Appendix. C.2.

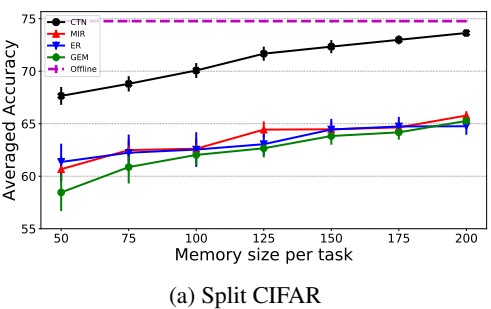 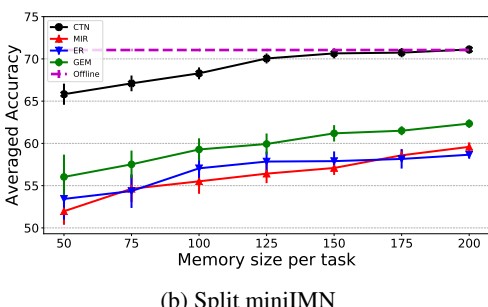

(a) Split CIFAR              (b) Split miniIMN

Figure 2: ACC(↑) as a function of the episodic memory size on the Split CIFAR-100 and Split miniIMN benchmarks. Best viewed in colors.

We use a multilayer perceptron with two hidden layers of size 256 for pMNIST, a reduced ResNet18 with three times fewer filters (Lopez-Paz & Ranzato, 2017) for Split CIFAR and Split miniIMN, and a full ResNet18 on CORE50. Following (Lopez-Paz & Ranzato, 2017), we use a Ring buffer as the memory structure for all methods and random sampling to select data from memory, including the episodic and semantic memories of CTN. The exceptions are MER (Riemer et al., 2019), which uses reservoir sampling, and MIR (Aljundi et al., 2019a), which use their sampling strategies as proposed by the authors. For CTN, the episodic memory and semantic memory are implemented as two Ring buffers with sizes equal to 80% and 20% of the total budget. This configuration is also cross-validated from the validation tasks. For each incoming batch of data, we randomly push 80% samples to the current task's episodic memory and the other 20% are for the current task's semantic memory.

We follow the procedure proposed in Chaudhry et al. (2019a) to cross-validate all hyperparameters using the first three tasks. Then, the best configuration is selected to perform continual learning on the remaining tasks. During continual learning, the task identifier is given to all methods. We optimize all models using SGD with a mini-batch of size ten over *one* epoch. We run each experiment five times, each has the same task order but different initialization seed, and report the following metrics: Averaged Accuracy (Lopez-Paz & Ranzato, 2017): ACC(↑) (higher is better) , Forgetting Measure (Chaudhry et al., 2018): FM(↓) (lower is better), and Learning Accuracy (Riemer et al., 2019): LA(↑) (higher is better).

## 4.2 Results of Continual Learning Benchmarks

Table 1 reports the evaluation metrics of the models on four continual learning benchmarks considered with 50 samples per task. We observe that CTN is even comparable with the independent method and outperforms other baselines by a large margin. We remind that the independent method has $T$ times more parameters than the remaining methods, where $T$ is the total number of tasks. Moreover, CTN can exploit the relationship across tasks via the task identifiers to improve its performance. For example, learning to classify "man" and "woman" may be helpful to classify "boy" and "girl" because they belong to the same superclass "people". Finally, CTN significantly outperforms the baselines by achieving a better trade-off between alleviating catastrophic forgetting and facilitating knowledge transfer, as shown by lower FM(↓) and higher LA(↑) . Overall, CTN achieves state-of-the-art results, even comparable with arge scale dynamic architecture method, while enjoying neglectable model complexity overhead compared to fixed architecture methods.

**ACC(↑) as a function of the episodic memory size.**

We study the models' performances as the memory size increases. We consider the Split CIFAR 100 and Split miniIMN benchmarks and train the models of CTN, ER, MIR, and GEM with the total memory size per task increasing from 50 to 200. Fig. 2 plots the ACC(↑) curves as a function of the memory size. Generally, the performances of all methods increase with larger memory sizes. Overall, CTN consistently outperforms the competitors across all memory sizes. Notably, in both benchmarks, CTN can achieve comparable performances to the Offline model even when the memory size per task is only 175. The results show that CTN not only excels in the low memory regime but also scales remarkably well when more memory budget is allowed.

Table 2: Evaluation metrics on the Small Split CIFAR benchmarks, M denotes the memory per task

| Method | Reduced Split CIFAR 25%, M = 50 | | | Reduced Split CIFAR 25%, M = 25 | | |
|---|---|---|---|---|---|---|
| | ACC(↑) | FM(↓) | LA(↑) | ACC(↑) | FM(↓) | LA(↑) |
| GEM | 51.01±0.95 | 5.65±1.09 | 53.39±0.98 | 47.33±0.89 | 8.77±1.58 | 53.79±1.35 |
| ER-Ring | 52.02±0.90 | 4.31±0.94 | 53.97±0.65 | 48.15±0.87 | 8.22±1.17 | 53.88±0.93 |
| MIR | 50.82±0.83 | 5.22±0.68 | 53.27±1.05 | 47.19±0.54 | 8.41±0.94 | 53.51±0.74 |
| CTN | **61.27±0.93** | **4.19±0.78** | **61.92±1.15** | **56.17±1.63** | **7.71±1.22** | **61.40±0.64** |
| Method | Reduced Split CIFAR 10%, M = 50 | | | Reduced Split CIFAR 10%, M = 25 | | |
| | ACC(↑) | FM(↓) | LA(↑) | ACC(↑) | FM(↓) | LA(↑) |
| GEM | 44.06±1.31 | 6.96±0.87 | 48.67±0.84 | 42.67±1.62 | 8.39±1.35 | 49.46±0.40 |
| ER-Ring | 44.60±1.65 | 6.07±1.77 | 48.36±0.46 | 43.09±1.22 | 7.68±1.94 | 49.40±1.40 |
| MIR | 46.63±0.56 | 4.38±0.45 | 48.35±0.52 | 44.12±0.94 | 6.84±1.05 | 48.48±0.76 |
| CTN | **56.61±0.74** | **4.33±0.48** | **58.77±0.99** | **52.64±0.63** | **6.74±0.73** | **57.27±1.02** |

Table 3: ACC(↑) of each component in CTN on Split CIFAR and Split mini Imagenet with 50 memory slots per task. BC: behavioral cloning (Eq. 6), C: controller, BO: Bilevel optimization (Eq. 3)

| | BC | C | BO | Split CIFAR | | | Split miniIMN | | |
|---|---|---|---|---|---|---|---|---|---|
| | | | | ACC(↑) | FM(↓) | LA(↑) | ACC(↑) | FM(↓) | LA(↑) |
| CTN | ✓ | ✓ | ✓ | **67.65±0.43** | **6.33±0.70** | 73.43±0.45 | **65.82±0.59** | **3.02±1.13** | 67.73±1.73 |
| | ✓ | ✓ | | 66.37±0.53 | 9.64±0.98 | **75.40±0.60** | 60.04±1.37 | 10.48±0.99 | **69.87±0.60** |
| | | ✓ | ✓ | 64.46±1.16 | 8.51±1.53 | 72.23±0.54 | 61.01±1.09 | 5.31±0.94 | 64.35±0.83 |
| | | ✓ | | 62.76±0.49 | 10.10±0.78 | 72.12±0.41 | 58.95±1.76 | 9.08±1.61 | 66.94±0.83 |
| ER | | | | 61.36±1.01 | 7.20±0.72 | 67.05±1.08 | 53.43±1.18 | 11.21±1.35 | 63.46±1.05 |

## 4.3 RESULTS ON LEARNING WITH LIMITED TRAINING DATA

One important goal of continual learning is to be able to learn with a limited amount of training data per task. This setting is much more challenging because it tests the learner's ability to quickly acquire knowledge only with limited training samples by utilizing its past experiences. In this experiment, we explore how different memory-based methods perform with only limited *training samples* per task and memory size. We consider the Split CIFAR benchmark; however, we reduce the amount of training data per task significantly. Particularly, we only consider 25% and 10% of the original data per task while the test data remains the same. We name the new benchmarks Reduced Split CIFAR 25% and Reduced Split CIFAR 10%, respectively. Notably, the Reduced Split CIFAR 10% only has **five samples** per class, which is extremely challenging. We compare CTN with GEM, ER, and MIR on these benchmarks with the memory size of 50 and 25 samples per task.

Table 2 shows the results of this experiment. When the training data are scarce, the baselines performances drop significantly, even below 50% ACC(↑) in three settings. CTN, on the other hand, consistently outperforms the baselines by a large margin, from **8% to 10%** across benchmarks, even in the challenging Reduced Split Cifar 10%. Moreover, the three baselines have similarly low LA, showing that they struggle in acquiring new knowledge when the training data of each task are limited. On the other hand, CTN can leverage information about the task-specific features to improve knowledge transfer and the learning outcomes. It is worth noting that even with 25% training data and 50 memory slots per task, CTN already outperforms several baselines that are trained with full data by cross-referencing the results with Table 1.

## 4.4 ABLATION STUDY

We study the contribution of each component in CTN in its overall performance and consider the Split CIFAR and Split miniIMN benchmarks with an episodic memory of 50 samples per task. Particularly, we are interested in how (1) the controller, (2) the bi-level optimization, and (3) the behavioral cloning strategy contribute to the base model. We implement variants of CTN with different combinations

Table 4: Model complexity of CTN with various backbone architectures

| Backbone | | | Controller | | Total | Increase |
|---|---|---|---|---|---|---|
| Structure | # Params | Structure | # Params | | | |
| MLP [784-256-256-10] | 269,322 | Linear model | 17,728 | 287,050 | 6.58% |
| ResNet18 (Lopez-Paz & Ranzato, 2017) | 1,095,555 | Linear model | 20,992 | 1,116,547 | 1.92% |
| ResNet18 (He et al., 2016) | 11,202,162 | Linear model | 59,200 | 11,261,362 | 0.53% |

Table 5: Averaged running time (in seconds) of compared methods on the task-aware continual learning benchmarks. All methods use M=50 memory slots per task, Ring buffer, and up to four gradient updates per samples

| Benchmark \ Method | ER-Ring | MIR | AGEM | CTN | GEM |
|---|---|---|---|---|---|
| pMNIST | 61 | 92 | 90 | 110 | 103 |
| Split CIFAR100 | 632 | 1030 | 680 | 910 | 1700 |
| Split miniIMN | 1320 | 2130 | 1700 | 1890 | 2850 |

of these components and report the results in Table 3. Notably, CTN with only the controller (C) is equivalent to training the base network and the controller using the vanilla experience replay approach. Despite this, the controller can offer significant improvements over ER: over 5% ACC(↑) in Split miniIMN. When the controller is optimized by our proposed bilevel optimization (C + BO), the performances are further improved, showing that our proposed bilevel objective achieves a better trade-off between alleviating forgetting and facilitating knowledge transfer. Lastly, the behavioral cloning strategy can help alleviate forgetting and further strengthen the results. Overall, each of the proposed components adds positive contributions to the base model, and they work collectively as a holistic method and achieved state-of-the-art results in continual learning.

### 4.5 COMPLEXITY ANALYSIS

In this section, we study the CTN's complexity with the backbones used in our experiments and report the results in Table 4. In all cases, the controller only adds minimal additional parameters, almost neglectable in complex deep architectures such as ResNets (He et al., 2016; Lopez-Paz & Ranzato, 2017). Therefore, we can safely compare CTN with other fixed architecture methods using the same backbone because they have nearly the same number of parameters.

Table 5 reports the averaged running time (in seconds) of considered methods. All methods are implemented using Pytorch (Paszke et al., 2019) version 1.5 and CUDA 10.2. Experiments are conducted using a single K80 GPU and all methods are allowed up to four gradients steps per sample. Clearly, ER-Ring has the most efficient time complexity thanks to its simplicity. On the other hand, GEM has high computational costs because of its quadratic constraints. MIR also exhibits high running time because of its virtual update, which doubles the total gradient updates. CTN, in general, is slightly faster MIR and more efficient than GEM. Overall, CTN achieves a great trade-off between model/computational complexity and performance: CTN's performances are significantly higher than considered baselines with only minimal memory and computational overhead.

## 5 CONCLUSION

In this work, we study the online continual learning problem and propose Contextual Transformation Networks (CTN), where a fixed architecture network can model both the common features and specific features of each task. CTN works by employing a controller that modifies features of the base network conditioning on the task identifiers. To optimize CTN, we further propose a novel dual memory system equipped with a bilevel optimization objective that can efficiently transfer knowledge and alleviate forgetting simultaneously. Moreover, we discuss the relationship of CTN to the Complementary Learning Systems theory in neuroscience and meta learning from different perspectives, showing that CTN is related to other disciplines. Through extensive experiments, our results demonstrate that CTN consistently outperforms fixed architecture methods and achieves state-of-the-art results. Moreover, CTN is even comparable with a large scale dynamic architecture network, while enjoying almost no additional model complexity.

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

APPENDIX

This Appendix is organized as follows. In Appendix A, we provide the details of the continual learning protocol and evaluation metrics used in this work. Appendix B provides pseudo-code of CTN and its implementation on standard deep learning architectures such as MLP and Residual networks. Appendix C provides additional experiment details, including the summary of our benchmarks, results of additional baselines, model and computational complexity, and hyperparameter settings.

## A  CONTINUAL LEARNING PROTOCOLS

Continual learning, a.k.a. lifelong learning, McCloskey & Cohen (1989); Thrun & Mitchell (1995); Ring (1997) has been extensive studied over the past decades. In this work, we consider the problem of online continual learning studied by (Lopez-Paz & Ranzato, 2017; Chaudhry et al., 2019a).

Specifically, at time step $t$, a learner receives an input pair $(\boldsymbol{x}, t)$ and makes a prediction $y = f(x, t; \boldsymbol{w})$ by a predictor $f(\cdot)$ parameterized some parameter $\boldsymbol{w}$. Note that here the input $\boldsymbol{x}$ belongs to an underlying task $\mathcal{T}_t$, which is also given to the learner. Each task $\mathcal{T}_t$ comprises a training dataset $\mathcal{D}_t^{tr}$, whose data will be sequentially presented to a learner, and a separate testing set $\mathcal{D}_t^{te}$. Following (Chaudhry et al., 2019a), we also assume having access to a small amount of tasks prior to learning for hyperparameter validation and an episodic memory $\mathcal{M}$ can be used. We assume that the stream of data $\{(\boldsymbol{x}_i, t_i), y_i\}_{i=1}^{\infty}$ arrives sequentially and the goal is to optimize a model that can perform well on all observed tasks so far.

To measure the model performance, we adopt three standard metrics: Average Accuracy ACC($\uparrow$) (Lopez-Paz & Ranzato, 2017), Forgetting Measure FM($\downarrow$) Chaudhry et al. (2019a), and Learning Accuracy LA($\uparrow$) (Riemer et al., 2019). Denote $a_{i,j}$ as the model's accuracy evaluated on the test set $\mathcal{D}_j^{te}$ after it has been trained on the most recent sample in dataset $\mathcal{D}_i$ of task $\mathcal{T}_i$. Then, the above metrics are defined as:

- **Average Accuracy (higher is better):**  the average accuracy of all observed tasks:

$$\text{ACC}(\uparrow) = \frac{1}{T} \sum_{i=1}^{T} a_{T,i}.$$

- **Forgetting Measure (lower is better):**  the average forgetting of all previous tasks:

$$\text{FM}(\downarrow) = \frac{1}{T-1} \sum_{j=1}^{T-1} \max_{l \in \{1, \dots T-1\}} a_{l,j} - a_{T,j}.$$

- **Learning Accuracy (higher is better):** measures the performance of a model on a task right after it finishes training that task:

$$\text{LA}(\uparrow) = \frac{1}{T} \sum_{i=1}^{T} a_{i,i}.$$

In the literature, there exists several different continual learning protocols. Here we categorize them based on two questions: (i) Is information about the task of a sample given during training and testing? (ii) Does training within each task is performed online? For question (i), when we do not know which task does the sample belong to, evaluation is called "single-head" and there is a shared classifier for all tasks (Aljundi et al., 2019b). In question (ii), data of a task can either be fully available when task changes or can arrives sequentially. When all the task data is available, training within tasks can be done in an offline fashion with multiple epochs through data. Our protocol used is in this work is proposed in Lopez-Paz & Ranzato (2017) in which data of each task arrives sequentially and task identifier is also given. Moreover, hyperparameter cross-validation is also an important problem in continual learning, regardless of the protocol considered. Particularly, we must not use data of future tasks when searching for the hyperparameter. Here we follow Chaudhry et al. (2019a) and assume that we have access to a small amount of tasks prior to continual learning. Such tasks will not be encountered again during actual continual learning and only be used for cross-validation.

# B IMPLEMENTING CONTEXTUAL TRANSFORMATION NETWORKS

## B.1 PSEUDO-CODE

We provide the details algorithm of our CTN and its subroutines in Alg. 1. For simplicity, we drop the dependency of the losses on the parameters and use $\mathcal{L}^{tr}(\mathcal{B}_n)$ to denote $\mathcal{L}^{tr}(\phi, \varphi, \mathcal{B}_n; \theta)$ and $\mathcal{L}(\mathcal{B}_n)$ to denote $\mathcal{L}(\phi, \varphi, \mathcal{B}_n; \theta)$

---

**Algorithm 1:** Contextual Transformation Networks (CTN)

1  **Algorithm** TrainCTN$(\theta, \phi, \mathcal{D}_{1:T}^{tr})$
    **Require:** base model $\phi$, controller $\theta$, classifier $\varphi$
    **Init:** $\theta, \phi, \varphi, \mathcal{M}_t^{em} \leftarrow \varnothing, \mathcal{M}_t^{sm} \leftarrow \varnothing$
2    **for** $t \leftarrow 1$ **to** $T$ **do**
3      **for** $j \leftarrow 1$ **to** $n_{batches}$ **do**           // Receive the dataset $D_t^{tr}$ sequentially
4        Receive a mini batch of data $\mathcal{B}_j$ from $\mathcal{D}_t^{tr}$
5        $\boldsymbol{x}^*, y^* \leftarrow$ Random sampling from $\mathcal{B}_j$      // Sampling for the semantic memory
6        $\mathcal{M}_t^{sm} \leftarrow$ **MemoryUpdate**$(\mathcal{M}_t^{sm}, \{\boldsymbol{x}^*, y^*\})$   // Update the semantic memory
7        $\mathcal{M}_t^{em} \leftarrow$ **MemoryUpdate**$(\mathcal{M}_t^{em}, \mathcal{B}_j)$      // Update the episodic memory
8        **for** $i \leftarrow 1$ **to** $n_{outer}$ **do**
9          **for** $n \leftarrow 1$ **to** $n_{inner}$ **do**
10            $\mathcal{B}^{em} \leftarrow$ Sample$(\mathcal{M}_{<t}^{em})$
11            $\mathcal{B}_n \leftarrow \mathcal{B}^{em} \cup \mathcal{B}_j$
12            $\phi \leftarrow \phi - \nabla_\phi \mathcal{L}^{tr}(\mathcal{B}_n)$      // Inner update the base model $\phi$
            $\varphi \leftarrow \varphi - \nabla_\varphi \mathcal{L}^{tr}(\mathcal{B}_n)$      // Inner update the classifier $\phi$
13         $\mathcal{B}^{sm} \leftarrow$ Sample $(\mathcal{M}_{\leq t}^{sm})$
14         $\theta \leftarrow \theta - \nabla_\theta \mathcal{L}(\mathcal{B}^{sm})$      // Outer update the controller $\theta$
15     $\mathcal{M}_t^{em} \leftarrow \mathcal{M}_t^{em} \cup \{\pi(\hat{y}/\tau)\}$      // Calculate the behavioural cloning outputs
16     $\mathcal{M}^{em} \leftarrow \mathcal{M}^{em} \cup \mathcal{M}_t^{em}$      // Update the total episodic memory
17    **return** $\theta, \phi$

1  **Procedure** Forward$(\theta, \phi, \varphi, \boldsymbol{x}, t)$
2    $\gamma_t, \beta_t \leftarrow c_\theta(t)$      // Calculate the transforming coefficients
3    $\tilde{h}(\boldsymbol{x}; t) \leftarrow \frac{\gamma_t}{\|\gamma_t\|_2} \otimes \hat{h}(\boldsymbol{x}) + \frac{\beta_t}{\|\beta_t\|_2}$      // Calculate the task-specific features
4    **return** $g_{\varphi_t}(h(\boldsymbol{x}, t))$

1  **Procedure** MemoryUpdate$(\mathcal{M}, \mathcal{B})$
    **Require:** Implement $\mathcal{M}$ as a queue (FIFO) data structure
2    **for** $(\boldsymbol{x}, y)$ **in** $B$ **do**
3      $\mathcal{M}$.append$(\boldsymbol{x}, y)$
4    **return** $\mathcal{M}$

---

## B.2 IMPLEMENTING CTN ON COMMON ARCHITECTURES

In this section, we provide the implementation details of CTN on two feedforward network bases that we use in our experiments. We implement the context model as a single regression layer. Moreover, we share the parameter of the scale and shift models $\gamma, \beta$, resulting in one set of parameters that takes a task embedding as input and outputs both scale and shift values for a particular layer of the base network. Next, we will describe our implementation of CTN with the base network as MLP and ResNet (He et al., 2016). For CTN, we will use $\hat{h}$ as the original features, $\tilde{h}$ as the task-specific features, and $h$ as the combine features.

**CTN with Multilayer Perceptron.** Consider an $L-$layers MLP with the form:

$$
\begin{aligned}
h_0 &= \boldsymbol{x} \\
h_l &= \text{ReLU}(\boldsymbol{W}_l^\top h_{l-1}), \forall l = 1, \ldots, L-1, \\
h_L &= g_t = \text{Softmax}(\boldsymbol{W}_{L,t}^\top h_{L-1})
\end{aligned}
$$

where the last layer is the softmax classifier $h_t$. Since the last classification layer is already conditioned on the task information, here we are interested in conditioning the intermediate layers $h_{l<L}$. The CTN with MLP is implemented as:

$$
\begin{aligned}
\hat{h}_0 &= \boldsymbol{x} \\
\hat{h}_l &= \text{ReLU}(\boldsymbol{W}_l^\top h_{l-1}), \forall l = 1, \ldots, L-1, \\
\tilde{h}_l &= \text{ReLU}(\gamma_t \otimes \boldsymbol{W}_l^\top h_{l-1} + \beta_t), \forall l = 1, \ldots, L-1, \\
h_l &= \hat{h}_l + \tilde{h}_l \\
h_L &= g_t = \text{Softmax}(\boldsymbol{W}_{L,t}^\top h_{L-1})
\end{aligned}
$$

We condition each hidden layer of a MLP by using one context network for each layer. Each context network does not share parameters, however, the scale and shift models for one layer is shared.

**CTN with Deep Residual Network.** Unlike MLP, we apply the task conditioning after the residual blocks instead of each convolution layer. Particularly, given a residual block defined as:

$$
\begin{aligned}
\hat{h}_1 &= \text{ReLU}(\text{BN}(\text{conv}(\boldsymbol{x}))) & \hat{h}_2 &= \text{BN}(\text{conv}(h_1)) \\
\hat{h}_3 &= \text{BN}(\text{conv}(\boldsymbol{x})) & \hat{h}_4 &= \text{conv}(x) \\
\bar{h} &= \hat{h}_3 + \hat{h}_4
\end{aligned}
$$

The task-conditioned residual block is computed as:

$$
\tilde{h} = \text{ReLU}(\bar{h}) + \text{ReLU}(\gamma_t \otimes \bar{h} + \beta_t)
$$

While in principle, it is possible to have a context network for each of the residual block, we empirically found that this does not offer significant improvements over using only one controller on the last residual block. Therefore, we only use one controller on the last residual block in all experiments that use a ResNet.

## C  EXPERIMENT DETAILS

### C.1  DATASET SUMMARY

We summary the datasets used in our experiments in Table 6.

Table 6: Summary of datasets used in our experiments

| Dataset | Classes | Train | Test | Dimension |
|---|---|---|---|---|
| MNIST (LeCun et al., 1998) | 10 | 1,000 | 10,00 | $28 \times 28$ |
| CIFAR100 (Krizhevsky & Hinton, 2009) | 100 | 50,000 | 10,000 | $3 \times 32 \times 32$ |
| miniIMN (Vinyals et al., 2016) | 100 | 50,000 | 10,000 | $3 \times 84 \times 84$ |
| CORe50 (Lomonaco & Maltoni, 2017) | 50 | 119,894 | 44,971 | $3 \times 84 \times 84$ |

For each benchmark, we normalize the pixel values to $[0, 1]$ by dividing their values by 255.0 as used in Lopez-Paz & Ranzato (2017), no other data preprocessing steps are performed.

### C.2  ADDITIONAL BASELINES

In Table 7, we provide a more comprehensive comparison with more baselines in the four benchmarks considered: Permuted MNIST, Split CIFAR100 and Split miniIMN, and CORe50. Some of these baselines are less competitive, thus, were not included in the main paper due to space constraints. We provide a brief descrption of each baselines in the following.

Table 7: Evaluation metrics on continual learning benchmarks considered. All methods use the same backbone network for all benchmarks, episodic memory size is M=50 samples per task

| Method | pMNIST | | | CORe50 | | |
|---|---|---|---|---|---|---|
| | ACC(↑) | FM(↓) | LA(↑) | ACC(↑) | FM(↓) | LA(↑) |
| Finetune | 61.66±1.50 | 20.67±1.64 | 80.89±0.45 | 4.38±0.10 | 49.66±1.14 | 49.08±1.20 |
| LwF | 63.31±3.56 | 14.29±3.05 | 75.76±1.43 | 31.20±0.66 | 20.44±1.37 | 49.20±1.10 |
| EWC | 67.34±3.00 | 11.00±2.36 | 76.59±1.49 | 31.86±3.90 | 14.34±3.08 | 42.98±2.50 |
| GEM | 74.84±0.95 | 8.57±0.33 | 81.74±0.77 | 42.56±0.86 | 7.36±0.90 | 46.84±2.22 |
| KDR | 72.97±0.58 | 9.20±0.44 | 81.40±0.41 | OOM | OOM | OOM |
| AGEM | 68.67±0.71 | 13.98±0.68 | 81.54±0.25 | 40.28±3.15 | 11.08±4.01 | 46.68±1.51 |
| MER | 76.59±0.74 | 6.88±0.59 | 82.09±0.33 | 39.28±1.25 | 9.08±1.25 | 45.52±0.96 |
| ER-Ring | 76.02±0.59 | 8.57±0.33 | 83.69±0.44 | 41.72±1.30 | 9.10±0.80 | 48.18±0.81 |
| MIR | 76.58±0.10 | 8.34±0.11 | 83.57±0.07 | 43.50±1.92 | 6.14±0.91 | 45.98±1.14 |
| BCL | 7.91±0.34 | 6.23±0.14 | 83.75±0.28 | 44.72±1.31 | 5.97±0.88 | 47.68±0.87 |
| CTN (ours) | **79.01±0.65** | **6.69±0.51** | **85.11±0.45** | **54.17±0.85** | **5.50±1.01** | **55.32±0.34** |
| Independent* | 81.05±0.29 | 0.00 | 81.05±0.29 | 53.54±1.10 | 0.00 | 53.54±1.10 |
| Offline | 84.95±0.95 | - | - | 89.73±0.91 | - | - |

| Method | Split CIFAR | | | Split miniIMN | | |
|---|---|---|---|---|---|---|
| | ACC(↑) | FM(↓) | LA(↑) | ACC(↑) | FM(↓) | LA(↑) |
| Finetune | 33.52±3.13 | 33.88±2.78 | 65.15±1.18 | 31.51±2.00 | 26.00±2.12 | 55.83±1.42 |
| EWC | 39.46±3.75 | 24.69±3.84 | 64.54±1.20 | 32.52±0.53 | 25.74±2.78 | 56.39±2.45 |
| ICARL | 50.27±0.84 | 16.55±0.82 | 65.83±1.53 | 44.95±0.08 | 17.59±0.40 | 61.46±0.50 |
| GEM | 57.77±0.86 | 10.93±1.03 | 66.45±0.06 | 55.04±1.88 | 7.81±1.70 | 60.13±1.36 |
| KDR | 62.75±0.80 | 5.01±0.79 | 66.11±0.70 | 56.89±2.45 | 4.83±1.23 | 59.29±1.31 |
| AGEM | 58.27±0.86 | 8.76±0.67 | 66.12±1.17 | 51.14±2.16 | 6.99±1.96 | 55.11±0.76 |
| MER | 61.32±0.86 | 11.90±0.86 | 72.51±0.41 | 57.94±1.08 | 8.98±0.79 | 66.11±0.76 |
| ER-Ring | 61.36±1.01 | 7.20±0.72 | 67.05±1.08 | 53.43±1.18 | 11.21±1.35 | 63.46±1.05 |
| MIR | 63.37±1.99 | 10.53±1.63 | 73.27±0.77 | 51.97±1.58 | 10.37±2.72 | 60.63±3.43 |
| BCL | 63.87±2.27 | **4.93±0.75** | 67.73±1.99 | 62.20±0.43 | 4.85±0.95 | 65.23±1.10 |
| CTN (ours) | **67.65±0.43** | 6.33±0.70 | **73.43±0.45** | **65.82±0.59** | **3.02±1.13** | **67.43±1.37** |
| Independent* | 67.21±0.51 | 0.00 | 67.21±0.51 | 65.85±0.98 | 0.00 | 65.85±0.98 |
| Offline | 74.11±0.66 | - | - | 71.15±2.95 | - | - |

- **Finetune**: a naive method that learns sequentially without any regularization.
- **LwF** (Li & Hoiem, 2017): prevents forgetting by a distillation loss of the previous model on current data.
- **EWC** (Kirkpatrick et al., 2017): penalizes the changes of important parameters to previous tasks to prevent forgetting.
- **GEM** (Lopez-Paz & Ranzato, 2017): uses an episodic memory to store some data and prevents the losses of old tasks from increasing during learning new tasks.
- **KDR** (Hou et al., 2018): uses knowledge distillation and task-specific experts to balance between learning new tasks and alleviating forgetting.
- **AGEM** (Chaudhry et al., 2019a): an efficient version of GEM by averaging the constraints in GEM.
- **MER**: (Riemer et al., 2019) maximizes the gradient inner product between every sample pair in the memory by a variant of the Reptile algorithm. We use MERAlg6 with mini batch of size 10 for consistency with remaining methods.
- **ER-Ring** (Chaudhry et al., 2019b): simply mixes data of previous and current tasks during training and optimizes a multitask loss.
- **MIR** (Aljundi et al., 2019a): is a variant of ER which selects the samples in the episodic memory that maximizes the model's forgetting to replay.

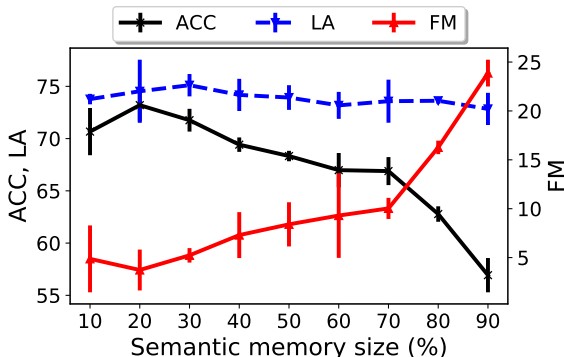

Figure 3: Effect of memory size on CTN's performance. For every semantic memory size $m \times 50$, the corresponding episodic memory size is $(1 - m) \times 50$.

- **BCL** (Pham et al., 2020): a bilevel-optimization method using Reptile update (Nichol et al., 2018) such that the base model can generalize to a separate memory units. Unlike BCL, our CTN can model the task-specific features and does not need approximations to solve the bilevel optimization problem.

- **Independent**[*] (Lopez-Paz & Ranzato, 2017): maintains a separate model for each task, each has the same number of parameters as other methods. While being unrealistic, this model was used as the upper bound model in Hung et al. (2019) thanks to its impressive performance.

- **Offline**: an upper bound model that performs multitask training on all data. Note that this model does not follow the continual learning setting. We implement the offline model by training the network three epochs over all data of all tasks.

### C.3 CTN Model Complexity and Computation Cost

**Model complexity.** Recall the interaction between the controller and the base model is described as:

$$\tilde{h}(\boldsymbol{x}; t) = \frac{\gamma_t}{\|\gamma_t\|_2} \otimes h(\boldsymbol{x}) + \frac{\beta_t}{\|\beta_t\|_2} \quad \text{and} \quad \{\gamma_t, \beta_t\} = c_{\boldsymbol{\theta}}(\boldsymbol{e}(t)), \tag{7}$$

where $h(\boldsymbol{x})$ is a feature map with dimension $(C \times H \times W)$ where $C$ is the number of channels, $H$ and $W$ are the spatial dimension of this feature map. A feature-wise affine transformation $\gamma_t, \beta_t$ is only required to have dimension $(C \times 1 \times 1)$ for each $\gamma_t$ and $\beta_t$. In our implementation, we predict both $\gamma_t$ and $\beta_t$ from the task embedding $\boldsymbol{e}(t)$ by a parameter $\boldsymbol{\theta}$. As a result, let $e$ be the task embedding dimension, the embedding layer will cost $(T \times \boldsymbol{e})$ and the controller (linear regression model) will cost $(2C \times e)$, resulting in the total $(T \times e + 2C \times e)$ parameters in the controller **for all tasks**. In practice, this term is dominated by $C \times e$ because $T$ and $\boldsymbol{e}$ are the number of tasks and the embedding dimension, which are quite small. When a new task arrives, we only need to allocate $e$ parameters in the embedding matrix.

Overall, CTN offers significantly performance improvements with only minimal memory overhead.

### C.4 Effect of The semantic memory Size

We study how the semantic memory size effects CTN performance. For this experiment, we consider the validation tasks in the Split CIFAR-100 benchmark (the first three tasks) and vary the semantic memory size and episodic memory size such that their total sizes equals to 50 samples per task.

Fig. 3 reports the results of this experiment. We can see that when the semantic memory size is 10 (20% of the total memory), CTN achieves the highest ACC, FM($\downarrow$) and lowest FM($\downarrow$) and these evaluation metrics degrades when the semantic memory sizes increases. Generally, we have to balance the amount of memory for controller and the base network. Since the controller is only a simple model, it only requires a small amount of data in the semantic memory.

C.5 Hyperparameter selection

We provide the hyper-parameters values of methods considered in our task-aware experiments. For brevity, we use MNIST to denote both the Permuted MNIST and Rotated MNIST benchmarks. The Small Split CIFAR experiments use the same hyper-parameter settings as the original Split CIFAR100. For each method, we use the same hyper-parameter notation and description as provided in the corresponding original papers.

- GEM
    - Learning rate: 0.03 (MNNIST, Split CIFAR100), 0.05 (Split miniIMN)
    - Gradient noise $\gamma$: 0.5 (all experiments)
    - Number of gradient updates: 1 (all experiments)
- AGEM
    - Learning rate: 0.03 (MNNIST), 0.1 (Split CIFAR100), 0.3 (Split miniIMN)
    - Number of to estimate gradient constraints: 1000 (MNIST), 850 (Splti CIFAR100, Split miniIMN)
    - Number of gradient updates: 1 (all experiments)
- MER
    - Learning rate: 0.03 (MNIST), 0.05 (Split miniIMN), 0.1 (Split CIFAR100)
    - Replay batch size: 64 (Permuted MNIST, Split CIFAR), 128 (Split miniIMN)
    - Number of gradient updates: 3 (all experiments)
    - Across batch leanring rate $\gamma$: 0.3 (all experiments)
- ER
    - Learning rate: 0.03 (MNIST, Split CIFAR100, Split mini Imagenet)
    - Replay batch size: 10 (all benchmarks)
    - Number of gradient updates: 3 (all experiments)
- MIR
    - Learning rate: 0.03 (MNIST, Split CIFAR100, Split mini Imagenet)
    - Replay batch size: 10 (all benchmarks)
    - Number of gradient updates: 3 (all experiments)
- CTN
    - Inner learning rate $\alpha$: 0.01 (all benchmarks)
    - Outer learning rate $\beta$: 0.05 (all benchmarks)
    - Regularization strength $\lambda$: 100 (all benchmarks)
    - Temperature $\tau$: 5 (all benchmarks)
    - Replay batch size: 64 (all benchmarks)
    - Number of inner and outer updates: 2 (all benchmarks)
    - Semantic memory size in percentage of total memory: 20% (all benchmarks)

Each hyper-parameter is cross-validated using grid search on the **three validation tasks**, which will not be encountered during continual learning. The grid for each hyper-parameter is provided below.

- Learning rate, including inner, outer (CTN) and across batch (MER) learning rates: $[0.01, 0.03, 0.05, 0.1, 0.3]$
- Number of gradient updates, including inner and outer updates (CTN): $[1, 2, 3, 4]$
- Replay batch size: $[10, 32, 64, 128]$
- Temperature $\tau$: $[1, 2, 5, 10]$
- Regularization strength $\lambda$ (CTN):

Table 8: Alternative strategies to reduce forgetting in CTN's inner optimization. BC: behavioural cloning strategy in Eq. 6

| Method | Split CIFAR | | | Split miniIMN | | |
|---|---|---|---|---|---|---|
| | ACC($\uparrow$) | FM($\downarrow$) | LA($\uparrow$) | ACC($\uparrow$) | FM($\downarrow$) | LA($\uparrow$) |
| CTN-BC | **67.65±0.43** | **6.33±0.70** | **73.43±0.45** | **65.82±0.59** | **3.02±1.13** | **67.73±1.73** |
| CTN-EWC | 60.33±1.44 | 9.33±1.55 | 68.78±0.24 | 57.69±0.96 | 5.59±0.45 | 61.53±1.38 |
| CTN-GEM | 64.40±2.52 | 8.06±1.92 | 71.49±0.46 | 60.65±0.80 | 5.83±0.84 | 64.42±0.46 |

- $\lambda$ (CTN): $[1, 10, 25, 50, 100]$
- $\gamma$ (GEM): $[0, 0.5, 1]$
- Semantic memory size in percentage of total memory (CTN): $[10\%, 20\%, 30\%, 40\%]$

## D    VARIANTS OF CTN

In this section, we explored alternative strategies for alleviating catastrophic forgetting in CTN's inner optimization problem, which is experience replay (ER) to train the base model $\phi$. Particularly, instead of the behavioural cloning strategy in Eq. 6, we consider two strategy to alleviate forgetting in ER by combining ER with EWC (Kirkpatrick et al., 2017) and GEM (Lopez-Paz & Ranzato, 2017). Table 8 show the results of this experiment on the Split CIFAR100 and Split miniIMN benchmarks. We can see that the behavioural cloning strategy significantly outperforms its competitors, EWC and GEM. Notably, using CTN with EWC requires larger episodic memory to store the previous tasks' parameters and their importance. Moreover, using CTN with GEM results in slower running time since GEM has the slowest training time as shown in Table 5. The results show that the behavioural cloning strategy is more suitable for alleviating forgetting in ER, while enjoying less memory overhead or faster running time compared to other alternatives.

