# OpenReview forum: "Contextual Transformation Networks for Online Continual Learning"
_ICLR.cc/2021/Conference — ICLR 2021 Poster_

### Official Review · AnonReviewer1 · 2020-10-19

**Rating:** 6
**Confidence:** 4

**Review:**

I have previously reviewed this submission for NeurIPS 2020 and cannot find substantial reason to alter my original review. I will therefore copy my original review along with the original rating.

However, as I do no longer have access to the file submitted by the authors for NeurIPS 2020, I encourage the authors to correct me by detailing modifications to the manuscript made for ICLR 2021, in which case I will be open to changing my score provided sufficient reason.

---

The authors introduce Contextual Transformation Networks (CTNs), a replay-based method for continual learning based on a dual-memory design and a controller that modulates the output of a shared based network to task-specific features.

Pros:
- Results are generally strong in comparison to other memory based CL techniques on accepted benchmarks.
- CTNs are a novel (albeit simple) architecture that might inspire future work.

Cons:
- The manuscript fails to describe the exact sampling strategies used for Semantic and Episodic Memory in sufficient details."semantic memory that stores a tiny set of data from all observed tasks" and "episodic memory that caches a small amount of past tasks’ training data". It's not clear to me what the difference between the two descriptions is. To me "observed" and "past" appear to be synonyms in this setting. Notation used later in the text uses no time index for the semantic memory (writing $M^{sm}$ as opposed to $M_t^{sm}$), indicating that the semantic memory remains fixed. If so, is the semantic memory initialised with data from all tasks at the beginning of training? Please describe this in more details.
- While the presented results on Image datasets are good, the CL community has to start considering more challenging and realistic tasks to make impact on other areas of Machine Learning. The perfect setting for "Online Continual Learning" is Reinforcement Learning and I was somewhat disappointed not to see this as an experiment.
- Baselines for this work are almost entirely focused on rehersal-based methods. It would have been nice to see modern Regularization or Dynamic architecture methods considered.
- The design of CTNs is somewhat ad-hoc and appears mainly based on heuristics and intuition with a rather loose neuroscience inspiration.
- Unfortunately I found the description of the proposed method somewhat confusing. Furthermore I suggest stating fundamental properties of the method more clearly.

---

Post-rebuttal: Raising my score from 5->6 after authors provided a detailed list of changes.

---

> ### Author Response · Authors · 2020-11-21
> **Detailed modifications to the manuscript made for ICLR 2021**
>
> Thank you for providing important feedback for our NeurIPS 2020 submission. We included your feedback and improved the manuscript substantially for our ICLR 2021 submission. Please find the detailed modifications compared to our NeurIPS 2020 submission below.
> - In Section 1, we clarified the motivation in developing CTN, which aims at developing a method that can achieve comparable performance to a strong dynamic architecture method, while enjoying minimal complexities: minimal additional parameters and running time, the memory is the same as other replay based methods (addressed concern #4: CTN’s design is somewhat adhoc).
>
> - In the main paper, we included the “Independent” baseline, which was a strong dynamic architecture method, and reported the results in Table 2. Notably, the “Independent” baseline was used as the upper bound model for the state-of-the art dynamic architecture method in Hung, Ching-Yi, et al. "Compacting, Picking and Growing for Unforgetting Continual Learning." NeurIPS. 2019. Due to space constraints, we provided the results of less competitive baselines, including EWC, in Table 7, Appendix C.2 (addressed concern #3: dynamic architecture baseline).
>
> - We changed the rotation MNIST (rMNIST) benchmark to the CORE50 benchmark, which was specifically designed for continual learning. The results on the CORE50 benchmark can be found in Table 2.
>
> - We included a complexity analysis experiment in Section 4.5 and reported the number of additional parameters and training time of all methods.
>
> - We clarified the description of the semantic memories in  the "Notations" and "Remark" paragraphs, Section 2. We fixed the notation issues, typos, and provided necessary background in Section 2 to improve the clarity of the proposed method (addressed concern #1: the semantic memory description and concern #5: the proposed method’s description is confusing).

---

> > ### Comment · AnonReviewer1 · 2020-11-23
> > **Reviewer response**
> >
> > I thank the authors for addressing my  concerns. I'm happy to raise my score to a 6.

---

### Official Review · AnonReviewer3 · 2020-10-24
**Well executed study, but approach is not very original**

**Rating:** 7
**Confidence:** 3

**Review:**

This paper tackles online continual neural network learning (following Lopez-Paz & Ranzato, 2017) with a combination of techniques: (1) a controller (or base parameter modulator, or hypernetwork) is introduced which produces task-specific scale and shift parameters, which modulate the feature maps of a base model (Perez et al., 2017); (2) as training data set samples are only provided once, the authors maintain experience replay buffers using soft label targets (by now standard techniques in online learning); (3) a bilevel optimization scheme where controller parameters are updated in an outer loop and base parameters adapted in an inner loop is employed. Task identities are available both during learning and at test time. This information is used to pick the correct task embeddings and the correct task-specific final classifier ("head").

The method achieves strong performance on appropriate benchmarks; hyperparameter tuning is handled carefully, which is not always the case in continual learning studies; the authors include a comparison to many relevant methods.

This paper's main weakness is low novelty. The method is essentially a careful application and combination of existing techniques. Some important references are missing or not correctly cited. In particular, the CNAPs method (Requeima et al., 2019) should be carefully discussed, and the differences to it highlighted. Also, the description of HAT (Serra et al.,2018) and task-conditioned hypernetworks (von Oswald et al., 2020) is not very accurate, as neither method requires "unbounded growth of the network"; in fact, both are quite related to the authors' approach, and both deserve more attention.

That being said, the experiments are well executed, and the study is comprehensive, as discussed above. I think that these merits outweigh the lack of novelty.

Some additional comments to the authors may be found below.
- The presentation of the method (which spans roughly two pages and a half) is lengthy and not very clear. For example the paragraph around eq. 1 is rather long, mixing related work with the actual method. There are a few imprecise statements, e.g., "We add an embedding layer on the task identifiers to avoid sparsity issues from using one-hot vectors". Is this embedding layer learned, and if so, how are sparsity issues avoided?

- Eq. 3 and SGD description below: in the outer problem, do the authors assume that $\phi^*$ is constant with respect to $\theta$? Here, CAVIA (Zintgraf et al., 2018) should be cited and discussed.

- The temperature hyperparameter $\tau$ is undefined in eq. 6.

- Should line 7 of the algorithm be omitted? If not, how are $B_M$ and $B_n$ defined?

- Conflicting notation issues: the authors use $\mathbf{c}_\theta(\cdot)$ to denote the controller (e.g., eq. 1), but also $\theta(t)$ (e.g., line 2 of procedure Forward, algorithm 1).

- Appendix B.2: I don't fully follow the equations below "CTN with Multilayer Perceptron". Why is $h_l$ obtained from $\tilde{h}_{l-1}$?

- The paper must be screened for typos and grammatical errors, there are plenty throughout.

---
Edit: after the authors' response I maintain my recommendation to accept the paper and keep my score of 7.

---

> ### Author Response · Authors · 2020-11-21
> **Response to Reviewer #3 Comments**
>
> We thank the reviewer for mentioning the missing related works and possible confusion in the dynamic architecture methods’ description. In the revision, included the missing references (Section 3.2) and discussed the works in Serra et al.,2018 and von Oswald et al., 2020 more carefully (Section 3.1). Regarding the motivation and novelty of our work, please see our respond to R4, concern #1.
>
> Regarding CNAPs and CAVIA, while both methods use FiLM, we want to highlight that CTN and CNAPs/CAVIA address different problems in different settings. Particularly, CNAPS and CAVIA are few-shot learning methods and they assume having access to a large amount of training tasks to obtain a good common feature representation. Given a new task with limited training data (e.g. 1 shot), the FiLM layers are used to quickly adapt the common knowledge to this task, balancing between over-fitting (adapting the whole network) and under-fitting (adapting only the classifier). In contrast, CTN learns both the common knowledge across tasks and the task-specific on the fly. Thus, CTN uses FiLM to model the task-specific features and benefits from the bilevel optimization to improve the learning of all observed tasks in the online setting. This difference is clearly highlighted in the continual learning experiment conducted in CNAPs: CNAPs initialization was a well-trained model; during continual learning, CNAPs froze the global parameters $\theta$ and only updated the adaptation network. On the other hand, CTN started with a random initialization and incrementally learned both the common and task-specific features.
>
> Concern #1: the presentation of the method is not clear, the embedding layer and sparsity issues. Thank you for your suggestion, we improved the presentation by moving the discussion to the Related Work section, leaving Section 2 focusing only on the proposed method.
> Regarding the sparsity issue, we consider the task identifiers as one-hot vectors, which are sparse and making training the controller directly on them difficult. Therefore, we introduce an additional embedding layer to embed one-hot vectors into dense vectors as input to the controller. The embedding layer is jointly trained with the controller. We clarified this in the revision.
>
> Concern #2 Eq.3 and the description below. In Eq. 3, the base model's parameters ($\phi^*$) are treated as constant, which follows the conventional bilevel optimization formulation: the inner/outer problem only optimizes the inner/outer variable. In the revision, we cited  CAVIA  in the related work.
>
> Concern #3: CTN’s equations in Appendix B.2. We apologize for this typo, the equations you mentioned are for the traditional MLP, and $\\tilde{h}_{l-1}$ should have been $h _ { l-1}$ instead.
>
> The temperature in Eq.6, line 7 in the Algorithm, conflicting notation uses. Thank you for carefully checking our draft, we fixed the typos and clarified the Algorithm’s details in the revision.

---

### Official Review · AnonReviewer4 · 2020-10-28

**Rating:** 6
**Confidence:** 3

**Review:**

This paper proposes CTN (Contextual Trasnformer Networks) for online continual learning. In particular, the authors introduce a dual memory framework that contains an episodic memory for base networks and semantic memory for task controllers. The overall framework is optimized with bi-level optimization. In addition, the base network also uses a KL-divergence loss to prevent catastrophic forgetting. Experiments are conducted on multiple datasets and the authors demonstrated that the proposed framewok outperforms other alternative approaches.

####### Strengths######
+ The paper is addressing an important problem, i.e., continual learning.
+ The motivation is clear.
+ Good results have been shown compared to other incremental learning methods.

#######Weakness######
- The novelty is a bit limited. The framework seems like a loose combination of existing well-explored techniques. For example, the modulation part  conditioned on tasks (Eqn 4) have been widely used before to modulate feature maps. For example.
[1] TAFE-Net: Task-Aware Feature Embeddings for Low Shot Learning
[2] TADAM: Task dependent adaptive metric for improved few-shot learning
The KL divergence to prevent forgetting has also been used before.

- The semantic memory and the  episodic memory are a bit confusing to me. What exactly are stored into the memory? Are image samples stored in the memory? From Algorithm 1,  L8 is updating the episodic memory with a batch of samples. But L16, suggests the predictions are added.

Minor:
+ Page 4 L3, the symbols are the same for the semantic memory and the  episodic memory.
+ Algorithm L8, the episodic memory is denoted as M^{tr} rather than M^{em}

########After rebuttal#########

I appreciate the author's effort in addressing my concerns. After reading the rebuttal and other reviews, I am raising my scores to 6.

---

> ### Author Response · Authors · 2020-11-21
> **Response to Reviewer  #4 Comments**
>
> We thank the Reviewer for your detailed review and questions. We address each comment individually.
>
> Concern #1: the novelty is limited. We respectfully disagree that CTN is a loose combination of existing works. Our main motivation is bridging the gap between two extremes of continual learning methods: dynamic architecture and fixed architecture methods. Therefore, we develop CTN, a holistic method that can efficiently learn task-specific features like a dynamic architecture method and achieve state-of-the-art performances, while enjoying the complexity of a fixed architecture method. We emphasize that CTN’s design is general and we choose FiLM as a specific implementation to keep the number of parameters comparable with other baselines. When more parameter budget is allowed, one can easily change FiLM to other feature transformations by adjusting the controller’s output dimension.
> Moreover, while FiLM has been used in other fields, naively applied it to continual learning with experience replay might not yield significant improvement. Therefore, we propose a novel bilevel optimization objective that can achieve a better trade-off between alleviating catastrophic forgetting and facilitating knowledge transfer, as shown in Table 3. We believe our findings are novel and important to continual learning, especially in the online setting where it is difficult to train deep networks to perform well (discussed in Section 1, first paragraph).
>
> Concern #2: the semantic memory description is confusing. Thank you for pointing out the possible confusion. The semantic memory stores the images and their corresponding labels. The episodic memory of a task stores 3 types of information: 1) the images, 2) the corresponding label, and 3) the soft labels obtained at the end of training that task. We clarified this in the revision.
>
> Minor concerns: typos in Page 4, L3 and Algorithm L8. Thank you for pointing out the typos, we fixed these in the revision.

---

### Official Review · AnonReviewer2 · 2020-10-29
**AnonReviewer2 Review**

**Rating:** 7
**Confidence:** 4

**Review:**

**Summary of paper**

This paper introduces a continual learning method called Contextual Transformation Networks (CTNs). CTNs consist of a base network and a controller, which outputs task-specific feature modulators. Both these have independent memories used to reduce forgetting. Additionally, the base network has an additional regularisation term. These two networks are trained together (formulated as a bi-level optimisation problem). Experiments on many different benchmarks are provided, with CTNs outperforming competing baselines on different metrics, often by large amounts. Some good additional experiments are also provided (different memory sizes, smaller datasets, ablations of the three major parts of CTNs).

**Review summary**

For me, the great experimental results/section means I am suggesting this paper be accepted. I do not see the intuition behind many of the design considerations in CTN: there is essentially a shared base network, with task-specific scaling/shifting (given by a controller network) and task-specific heads; a key difference to previous works is the use of two different memories for the two networks. But CTN has very strong performance in continual learning benchmarks.

**Pros of paper**

1. The strongest part of this paper for me is the experiments section. The results are extremely good and consistent over many different benchmarks. The authors provide different metrics for continual learning (not just average accuracy), and compare against some strong baselines. The additional experiments in Tables 3 and 4 are also very nice to see.
2. The method is presented well in Section 2, and Table 1 (comparing number of additional parameters) is nice.

**Cons of paper**

3. There are related works that I think the authors can mention, which have some similar ideas as in CTN (although all the ideas are never all put together as in CTN):
(a) FiLM layers for meta-learning / continual learning / multi-task learning:
[1] Requeima et al., 2019, "Fast and Flexible Multi-Task Classification using Conditional Neural Adaptive Processes"
[2] Loo et al. 2020, "Combining Variational Continual Learning with FiLM Layers"
[3] Rebuffi et al., 2018, "Efficient parametrization of multi-domain deep neural networks"
(b) Soft targets for continual learning
[4] van de Ven and Tolias, 2019, "Generative replay with feedback connections as a general strategy for continual learning"
4. How long does it take to perform the bilevel optimisation in CTN (can the authors provide training times ideally, or else complexity analysis)?
5. Section 3.1: "Dynamic architecture approaches... suffer from the unbounded growth of the network size and extensive resource usage". Is this true for all methods? Don't some methods try and achieve sublinear growth in parameters? Doesn't CTN also grow in a similar way?


**Additional comments/suggestions**

6. I found Figure 1 more difficult to understand than I think it needs to be. I think it would be useful for the authors to think how they can simplify it further for a future version of the paper.
7. Did the authors try some other continual learning algorithms to avoid forgetting in the base network other than "behavioral cloning"? (Eg EWC or GEM or any other method.)

**Update to review**

I shall keep my rating at 7. The authors updated the paper and I think it is good enough to be accepted. I still note how Figure 1 is difficult to parse (what are all the variables? Why are there so many arrows and boxes? It is not possible to understand what is going on without reading the text in detail, by which time it does not add much to the reader's understanding); in my opinion it requires starting from scratch again.

---

> ### Author Response · Authors · 2020-11-21
> **Reponses to Reviewer #2 Comments**
>
> Thank you for insightful comments and interesting suggestions to improve our work. We address your concern individually.
> Concern #3: missing related works. Thank you for suggesting related works that used FiLM, we already updated our related work section to discuss these works in the revision.
>
> Concern #4: training time.  We directly used the autograd function in Pytorch for the bilevel optimization in CTN and reported the running time for all experiments in Table 5, Section 4.5. Thanks to the small size of the controller, CTN's training time is in the middle of ER and GEM and is slightly faster than MIR.
>
> Concern #5: parameter growth in dynamic architecture methods. Thank you for pointing out the possible confusion. In that paragraph, we wanted to highlight that many dynamic architecture methods suffered from the growth of the backbone network. In contrast, the backbone in CTN was always fixed throughout training. Regarding the parameter growth, all CL methods, both dynamic and fixed architecture, have to allocate a new classifier when a new task arrives. We amended the writing accordingly to avoid future misunderstanding.
>
> Concern #6: improve Figure 1. Thank you for your suggestion, we simplified Figure 1 in the revision.
>
> Concern #7 other strategies to alleviate forgetting. Thank you for your interesting suggestion. We implemented two variants of CTN using EWC or GEM instead of behavioural cloning and reported the results in Table 8, Appendix D. The results show that while both methods can help alleviate forgetting in CTN, they are not as effective as the behavioural cloning strategy, which takes advantage of the additional information from soft labels.

---

### Author Response · Authors · 2020-11-21
**General Replies to All Reviewers**

We thank the reviewers for insightful comments and positive feedback. We are delighted that they found our method to be inspiring and well motivated (R1, R4). They agree that our experiments are extensive and the results are strong, impressive, and comprehensive (R1,R2,R3,R4). We amended our draft according to the Reviewers’ comments and will incorporate all feedback in the final version. We highlight the major changes from the current revision in the following.
- In the related work section, we added and discussed existing works that used FiLM or soft labels (R2). We also corrected the description of several dynamic architecture methods as suggested by R3.
- We added a new complexity analysis section (Section 4.5), which provided the running time of CTN in our experiments. We also moved the parameter analysis into Section 4.5 (R2).
- We fixed the possible confusion regarding the episodic memory, Algorithm 1 (R4), and typos pointed out by R3 and R4.
- We added two variants of CTN that use EWC and GEM instead of the behavioural cloning strategy to alleviate forgetting in Appendix D, Table 8(R2).

---

### Decision · Program_Chairs · 2021-01-07
**Final Decision**

**Decision:**

Accept (Poster)

**Comment:**

This work develops a novel framework for online continual learning, which they authors name Contextual Transformation Networks (CTN). This framework comprises a base network, which learns to map inputs to a shared feature representation, and a controller that efficiently transforms this shared feature vector to task specific features given a task identifier. Both of these components have access to their own memory. The optimization of the both the controller and base network parameters is framed as a bi-level optimization framework.

Pros:
- important and challenging problem
- strong results

Cons:
- Currently the writing creates the impression of limited novelty from a technical perspective. I would encourage the authors to more crisply highlight the technical novelty of their method.